Microbiology
Spectrum
| Open Peer Review | Clinical Microbiology | New-Data Letter

# Resolving a diagnostic challenge: first PCR-based detection of *Legionella longbeachae* in Italy

Silvia Guerriero,[1] Marilena La Sorda,[2] Brunella Posteraro,[3] Flavio De Maio,[2] Sara Cardinali,[2] Pierluigi Del Vecchio,[4] Giovanni Addolorato,[4] Claudia Tarli,[4] Rita Murri,[1,4] Massimo Fantoni,[4] Maurizio Sanguinetti,[2] Carlo Torti[1,4]

**KEYWORDS** *Legionella longbeachae*, antigen testing, PCR testing

The diagnosis of *Legionella* pneumonia, primarily a cause of community-acquired pneumonia, traditionally relies on urinary antigen testing, culture, and serology, all of which have limitations in detecting non-*Legionella. pneumophila* species (1). *Legionella longbeachae*, an emerging global cause of *Legionella* pneumonia (2), is often underdiagnosed unless molecular assays such as polymerase chain reaction (PCR) are applied (3).

Here, we report the first identification of *L. longbeachae* in our Italian center using a CE-marked real-time PCR assay, the Respiratory Bacterial ELITe MGB Panel assay. This multiplex PCR assay detects *Mycoplasma pneumoniae*, *Chlamydia pneumoniae*, and *L. pneumophila*/*L. longbeachae* DNA in respiratory samples (https://www.elitechgroup.com/molecular-diagnostics-intl/product/respiratory-bacterial-elite-mgb-panel/), adding to established NAAT assays (4–7).

An 81-year-old male with chronic obstructive pulmonary disease complicated by chronic respiratory failure, and a history of arterial hypertension, cerebral vasculopathy, and a previous multifocal pneumonia of unknown etiology, was admitted with fever (38°C), cough, confusion, and acute worsening of respiratory failure. He denied exposure to soil or potting compost. Hyponatremia and diffuse ground-glass opacities with evolving lobar consolidations on high-resolution CT scan raised suspicion of *Legionella* pneumonia.

Serologies for *Chlamydia* and *Mycoplasma* were negative. Urine antigen testing for *L. pneumophila* serogroups 1, 3, 5, 6, and 8 (SD Biosensor STANDARD F *Legionella* Ag FIA test) was negative. However, the SSI Diagnostica A/S ImmuView *L. pneumophila* and *L. longbeachae* urinary antigen test (8) yielded a positive result for *L. longbeachae* and a negative result for *L. pneumophila* serogroup 1. Bacterial cultures from a sputum sample, including those on *Legionella*-selective buffered charcoal yeast extract (BCYE) medium (9), yielded growth of *Klebsiella pneumoniae* but were negative for *Legionella* species. PCR performed on the same sputum sample using the ELITe MGB Panel assay detected *L. pneumophila*/*L. longbeachae*, whereas the FDA-cleared FilmArray Pneumonia Panel assay, which only detects *L. pneumophila* (4), was negative. Positive controls included *L. pneumophila* ATCC 33823 and *L. longbeachae* ATCC 33462.

Based on positive *L. longbeachae* results (illustrated in Fig. 1), intravenous levofloxacin (750 mg daily) was started, a first-line therapy—along with azithromycin—for *Legionella* pneumonia (1). Fluoroquinolones, such as levofloxacin, show higher *in vitro* activity against *L. longbeachae* clinical isolates compared to macrolides (10). Therefore, choosing fluoroquinolones over azithromycin could be advantageous when *L. longbeachae* (rather than *L. pneumophila*) is the causative agent in pneumonia cases. The patient showed progressive clinical and radiological improvement, and antibiotic therapy was discontinued after 9 days.

**Peer Reviewer** Portia Mira, Mayo Clinic, Rochester, Minnesota, USA

Address correspondence to Maurizio Sanguinetti, maurizio.sanguinetti@unicatt.it.

The authors declare no conflict of interest.

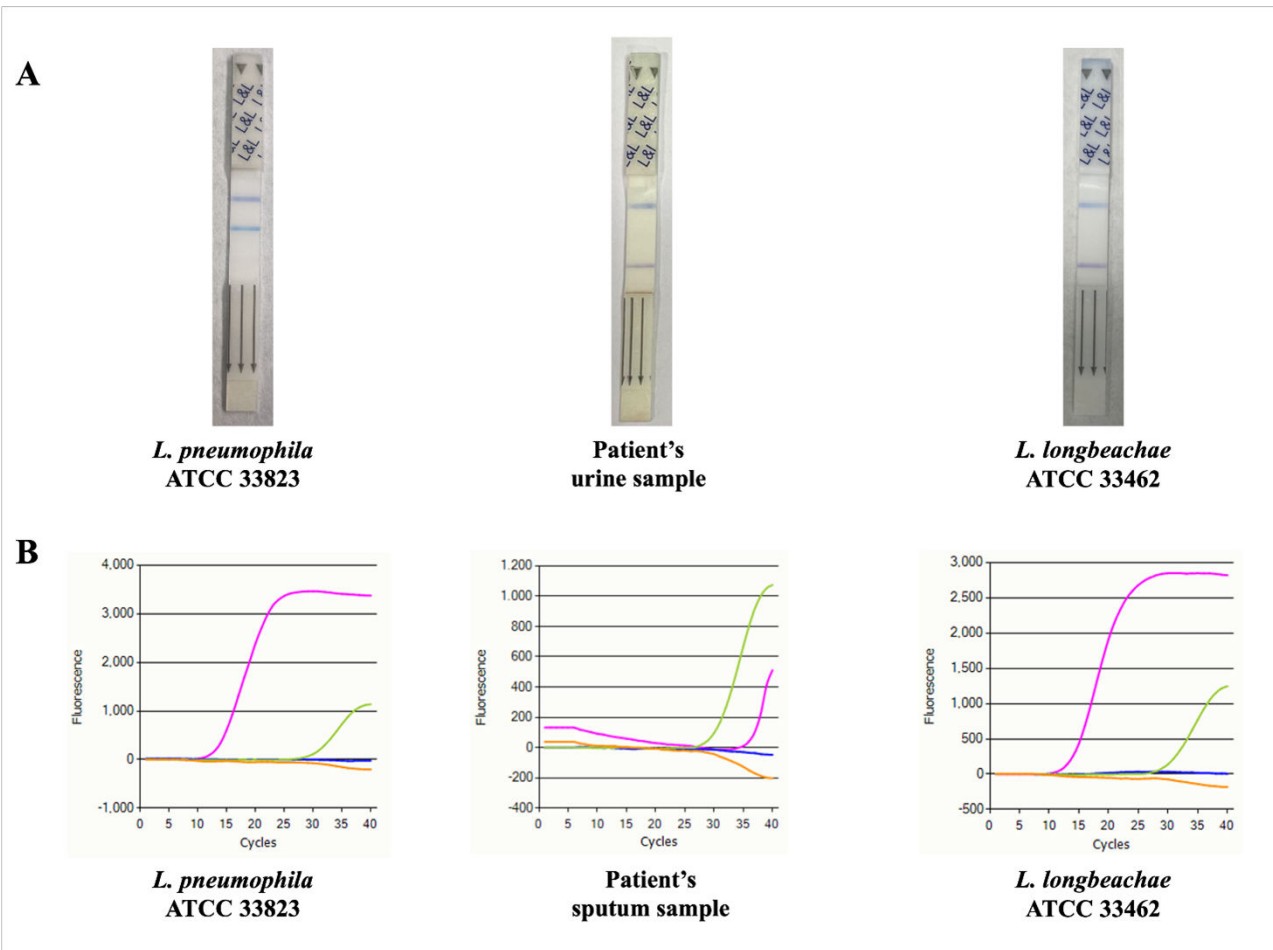

**FIG 1** Positive *L. longbeachae* detection results from the pneumonia patient's samples. Results are shown alongside those obtained in parallel with *L. longbeachae* ATCC 33462 and *L. pneumophila* ATCC 33823 reference strains, which served as positive controls. (A) The urine sample was analyzed using the SSI Diagnostica A/S ImmuView *L. pneumophila* and *L. longbeachae* urinary antigen test, yielding a *L. longbeachae*-specific immunochromatographic band. A similar band was visible in the *L. longbeachae* positive control, while no *L. pneumophila*-specific immunochromatographic band—detected in the positive control—was observed. (B) The sputum sample was analyzed using the Respiratory Bacterial ELITe MGB Panel assay, yielding a *L. pneumophila*/*L. longbeachae*-positive amplification curve (pink), crossing the fluorescence signal baseline at a threshold cycle (Ct) of 35.6, below the positivity threshold of 40 cycles. The internal control was amplified at a Ct of 28.3 (green), while no amplification was detected for *Chlamydia pneumoniae* (blue) or *Mycoplasma pneumoniae* (orange). Similar positive amplification curves were obtained with both *L. pneumophila* and *L. longbeachae* reference strains, with Ct values of 14.0 and 15.0, respectively (pink).

This case highlights the role of PCR testing, particularly NAAT assays that detect *Legionella* species beyond *L. pneumophila*, as a reliable alternative to BCYE culture when patients with community-acquired pneumonia test negative for *L. pneumophila* urinary antigen. Importantly, urinary antigen tests can remain positive even after pathogen clearance and treatment, while PCR provides higher sensitivity and specificity by directly detecting bacterial DNA, thereby offering a better correlation with active infection. While *L. longbeachae* culture methods require improvement (11), confirmation of PCR results was unlikely in our case due to the relatively high Ct value, suggesting a low bacterial load and reduced probability of culture positivity.

## AUTHOR AFFILIATIONS

[1]Dipartimento di Sicurezza e Bioetica, Università Cattolica del Sacro Cuore, Rome, Italy
[2]Dipartimento di Scienze di Laboratorio ed Ematologiche, Fondazione Policlinico Universitario A. Gemelli IRCCS, Rome, Italy

³Unità Operativa "Medicina di Precisione in Microbiologia Clinica," Direzione Scientifica, Fondazione Policlinico Universitario A. Gemelli IRCCS, Rome, Italy
⁴Dipartimento di Scienze Mediche e Chirurgiche, Fondazione Policlinico Universitario A. Gemelli IRCCS, Rome, Italy

## AUTHOR ORCIDs

Brunella Posteraro ⓘ http://orcid.org/0000-0002-1663-7546
Maurizio Sanguinetti ⓘ http://orcid.org/0000-0002-9780-7059

## AUTHOR CONTRIBUTIONS

Silvia Guerriero, Investigation, Methodology, Writing – review and editing | Marilena La Sorda, Investigation, Methodology, Writing – review and editing | Brunella Posteraro, Writing – review and editing, Data curation, Writing – original draft | Flavio De Maio, Investigation, Software, Writing – review and editing | Sara Cardinali, Investigation, Methodology | Pierluigi Del Vecchio, Investigation, Methodology | Giovanni Addolorato, Supervision | Claudia Tarli, Investigation, Methodology | Rita Murri, Investigation, Methodology, Writing – review and editing | Massimo Fantoni, Supervision, Writing – review and editing | Maurizio Sanguinetti, Funding acquisition, Project administration, Writing – review and editing | Carlo Torti, Supervision, Writing – review and editing

## ADDITIONAL FILES

The following material is available online.

### Open Peer Review

**PEER REVIEW HISTORY (review-history.pdf).** An accounting of the reviewer comments and feedback.

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
