## [Reviewer comments · Microbiology Spectrum]

Microbiology Spectrum

Resolving a Diagnostic Challenge: First PCR-Based Detection of *Legionella longbeachae* in Italy

Silvia Guerriero, Marilena La Sorda, Brunella Posteraro, Flavio De Maio, Sara Cardinali, Pierluigi Del Vecchio, Giovanni Addolorato, Claudia Tarli, Rita Murri, Massimo Fantoni, Maurizio Sanguinetti, and Carlo Torti

Corresponding Author(s): Maurizio Sanguinetti, Fondazione Policlinico Universitario Agostino Gemelli IRCCS

Review Timeline:

Submission Date:	March 21, 2025
Editorial Decision:	April 23, 2025
Revision Received:	April 29, 2025
Accepted:	May 26, 2025

Editor: Kenneth Gavina

Reviewer(s): Disclosure of reviewer identity is with reference to reviewer comments included in decision letter(s). The following individuals involved in review of your submission have agreed to reveal their identity: Portia Mira (Reviewer #1)

Transaction Report:

DOI: <https://doi.org/10.1128/spectrum.00860-25>

Re: Spectrum00860-25 (**Resolving a Diagnostic Challenge: First PCR-Based Detection of *Legionella longbeachae* in Italy**)

Dear Prof. Maurizio Sanguinetti:

Thank you for the privilege of reviewing your work. The work has been reviewed by two experts in this field and the decision is for re-submission with modifications. I would strongly encourage you to review the ASM Spectrum formatting information for New-data letters, particularly the word limit and requirements. Below you will find the full instructions from the Spectrum editorial office, as well as the reviewer comments.

Revision Guidelines

Sincerely,
Kenneth Gavina
Editor
Microbiology Spectrum

Reviewer #1 (Comments for the Author):

Summary:

This New Data Letter reports the first documented PCR-based detection of *Legionella longbeachae* in Italy using a CE-marked multiplex PCR assay (Respiratory Bacterial ELITe MGB Panel). The case involves an 81-year-old man with community-acquired pneumonia and negative urinary antigen results for *L. pneumophila*. Diagnosis was confirmed via PCR and a *L. longbeachae*-

specific urinary antigen test. Culture failed to grow Legionella, highlighting limitations of traditional diagnostics and the clinical utility of nucleic acid amplification testing (NAAT) for detecting non-pneumophila Legionella species.

This New Data Letter is clearly novel and clinically relevant. The authors have a well written and well-organized letter that demonstrates excellent understanding of the diagnostic limitations of current tests. They also include multiple diagnostic methods (culture, PCR, antigen, etc.) that provide a great comparison.

Areas for Improvement based on ASM New Data Letter Criteria:

Note that the ASM Spectrum "New Data Letter" format is brief, data-focused, and concise, emphasizing novel findings without extended discussion.

Comment 1: Length and Conciseness: The manuscript is a bit too long and reads more like a full case report or original research article. A recommendation is to condense to ~500 words max (ASM guideline). Focus on the core new data-the novel PCR-based detection of *L. longbeachae*-with minimal clinical background or discussion.

Comment 2: Much of the manuscript describes the patient's clinical course in detail, which is not the main novelty. My recommendation would be to shift emphasis from the patient history to the PCR assay's utility, confirmation using urinary antigen testing, lack of growth in culture, and the diagnostic implications. The full diagnostic timeline and detailed labs may not be necessary. A summary of the clinical information in a single short paragraph, focusing only on data relevant to the diagnostic challenge and assay performance.

Comment 3: There are background-heavy sections (e.g., on diagnostics in New Zealand) that read like a literature review. I'd suggest to trim these to 1-2 concise references that set context only, if necessary, this can help cut down on the length.

Comment 4: Table 1 and Figure 1 are valuable but may be too detailed for a Letter. Consider summarizing Table 1 more concisely or moving full details to Supplementary Material, if allowed. Keep only one key image in the main text.

Reviewer #2 (Comments for the Author):

One major suggestion: add in a comment/paragraph about how the PCR would be beneficial over the urinary antigen specific to *L. longbeachae*. I.e. comment about how long the urinary antigen test will be positive even after treatment etc.

Spectrum Review _ Spectrum00860-25

Corresponding author: Sanguinetti

Summary:

This New Data Letter reports the first documented PCR-based detection of *Legionella longbeachae* in Italy using a CE-marked multiplex PCR assay (Respiratory Bacterial ELITe MGB Panel). The case involves an 81-year-old man with community-acquired pneumonia and negative urinary antigen results for *L. pneumophila*. Diagnosis was confirmed via PCR and a *L. longbeachae*-specific urinary antigen test. Culture failed to grow *Legionella*, highlighting limitations of traditional diagnostics and the clinical utility of nucleic acid amplification testing (NAAT) for detecting non-*pneumophila* *Legionella* species.

This New Data Letter is clearly novel and clinically relevant. The authors have a well written and well-organized letter that demonstrates excellent understanding of the diagnostic limitations of current tests. They also include multiple diagnostic methods (culture, PCR, antigen, etc.) that provide a great comparison.

Areas for Improvement based on ASM New Data Letter Criteria:

Note that the ASM *Spectrum* "New Data Letter" format is brief, data-focused, and concise, emphasizing novel findings without extended discussion.

Comment 1: Length and Conciseness: The manuscript is a bit too long and reads more like a full case report or original research article. A recommendation is to condense to ~500 words max (ASM guideline). Focus on the core new data—the novel PCR-based detection of *L. longbeachae*—with minimal clinical background or discussion.

Comment 2: Much of the manuscript describes the patient's clinical course in detail, which is not the main novelty. My recommendation would be to shift emphasis from the patient history to the PCR assay's utility, confirmation using urinary antigen testing, lack of growth in culture, and the diagnostic implications. The full diagnostic timeline and detailed labs may not be necessary. A summary of the clinical information in a single short paragraph, focusing only on data relevant to the diagnostic challenge and assay performance.

Comment 3: There are background-heavy sections (e.g., on diagnostics in New Zealand) that read like a literature review. I'd suggest to trim these to 1–2 concise references that set context only, if necessary, this can help cut down on the length.

Comment 4: Table 1 and Figure 1 are valuable but may be too detailed for a Letter. Consider summarizing Table 1 more concisely or moving full details to Supplementary Material, if allowed. Keep only one key image in the main text.

Response to Reviewers

Manuscript ID: Spectrum00860-25

Title: *Resolving a Diagnostic Challenge: First PCR-Based Detection of Legionella longbeachae in Italy*

Reviewer #1 (Comments for the Author):

Summary: This New Data Letter reports the first documented PCR-based detection of Legionella longbeachae in Italy using a CE-marked multiplex PCR assay (Respiratory Bacterial ELITe MGB Panel). The case involves an 81-year-old man with community-acquired pneumonia and negative urinary antigen results for L. pneumophila. Diagnosis was confirmed via PCR and a L. longbeachae-specific urinary antigen test. Culture failed to grow Legionella, highlighting limitations of traditional diagnostics and the clinical utility of nucleic acid amplification testing (NAAT) for detecting non-pneumophila Legionella species.

This New Data Letter is clearly novel and clinically relevant. The authors have a well written and well-organized letter that demonstrates excellent understanding of the diagnostic limitations of current tests. They also include multiple diagnostic methods (culture, PCR, antigen, etc.) that provide a great comparison. Areas for Improvement based on ASM New Data Letter Criteria:

Note that the ASM Spectrum "New Data Letter" format is brief, data-focused, and concise, emphasizing novel findings without extended discussion.

Comment 1: Length and Conciseness

The manuscript is a bit too long and reads more like a full case report or original research article. A recommendation is to condense to ~500 words max (ASM guideline). Focus on the core new data-the novel PCR-based detection of L. longbeachae-with minimal clinical background or discussion.

Response: We have condensed the manuscript to approximately 500 words by summarizing the clinical course and removing nonessential background information.

Comment 2: Clinical Detail

Much of the manuscript describes the patient's clinical course in detail, which is not the main novelty. My recommendation would be to shift emphasis from the patient history to the PCR assay's utility, confirmation using urinary antigen testing, lack of growth in culture, and the diagnostic implications. The full diagnostic timeline and detailed labs may not be necessary. A summary of the clinical information in a single short paragraph, focusing only on data relevant to the diagnostic challenge and assay performance.

Response: We have substantially reduced the clinical details and placed greater focus on the PCR-based detection of L. longbeachae, the role of urinary antigen testing, and the diagnostic implications.

Comment 3: Background Review

There are background-heavy sections (e.g., on diagnostics in New Zealand) that read like a literature review. I'd suggest to trim these to 1-2 concise references that set context only, if necessary, this can help cut down on the length.

Response: We have significantly condensed the background information, retaining only the essential context for L. longbeachae underdiagnosis.

Comment 4: Figures and Tables

Table 1 and Figure 1 are valuable but may be too detailed for a Letter. Consider summarizing Table 1 more concisely or moving full details to Supplementary Material, if allowed. Keep only one key image in the main text.

Response: We have removed Table 1 from the main manuscript to comply with the New-Data Letter format. Figure 1 has been retained as the single key figure illustrating diagnostic results.

Reviewer #2 (Comments for the Author):

One major suggestion: add in a comment/paragraph about how the PCR would be beneficial over the urinary antigen specific to L. longbeachae. I.e. comment about how long the urinary antigen test will be positive even after treatment etc.

Response: We have added a comment to the final discussion paragraph, emphasizing that PCR testing provides higher sensitivity and specificity and better correlates with active infection compared to urinary antigen testing, which can remain positive after pathogen clearance.

Re: Spectrum00860-25R1 (**Resolving a Diagnostic Challenge: First PCR-Based Detection of *Legionella longbeachae* in Italy**)

Dear Prof. Maurizio Sanguinetti:

Please accept my sincerest apologies for the delay in the review of your revised manuscript, I have been travelling out of the country for work with limited access to internet - thank you for your patience.

Your manuscript has been accepted, and I am forwarding it to the ASM production staff for publication. Your paper will first be checked to make sure all elements meet the technical requirements. ASM staff will contact you if anything needs to be revised before copyediting and production can begin. Otherwise, you will be notified when your proofs are ready to be viewed.

Sincerely,
Kenneth Gavina
Editor
Microbiology Spectrum